# Nanopore metagenomic sequencing for detection and characterization of SARS-CoV-2 in clinical samples

**Nick P. G. Gauthier**[1], **Cassidy Nelson**[2], **Michael B. Bonsall**[2], **Kerstin Locher**[3,4], **Marthe Charles**[3,4], **Clayton MacDonald**[3,4], **Mel Krajden**[4,5], **Samuel D. Chorlton**[4,6]☉, **Amee R. Manges**[5,7]☉*

**1** Department of Microbiology and Immunology, University of British Columbia, Vancouver, British Columbia, Canada, **2** Mathematical Ecology Research Group, Department of Zoology, University of Oxford, Oxford, United Kingdom, **3** Division of Medical Microbiology, Vancouver General Hospital, Vancouver, British Columbia, Canada, **4** Department of Pathology and Laboratory Medicine, University of British Columbia, Vancouver, British Columbia, Canada, **5** British Columbia Centre for Disease Control, Vancouver, British Columbia, Canada, **6** BugSeq Bioinformatics Inc, Vancouver, British Columbia, Canada, **7** School of Population and Public Health, University of British Columbia, Vancouver, British Columbia, Canada

☉ These authors contributed equally to this work.
* amee.manges@ubc.ca

**Data Availability Statement:** Raw FASTQ data has been uploaded to NCBI Bioproject Accession PRJNA752146.

## Abstract

### Objectives

The COVID-19 pandemic has underscored the need for rapid novel diagnostic strategies. Metagenomic Next-Generation Sequencing (mNGS) may allow for the detection of pathogens that can be missed in targeted assays. The goal of this study was to assess the performance of nanopore-based Sequence-Independent Single Primer Amplification (SISPA) for the detection and characterization of SARS-CoV-2.

### Methods

We performed mNGS on clinical samples and designed a diagnostic classifier that corrects for barcode crosstalk between specimens. Phylogenetic analysis was performed on genome assemblies.

### Results

Our assay yielded 100% specificity overall and 95.2% sensitivity for specimens with a RT-PCR cycle threshold value less than 30. We assembled 10 complete, and one near-complete genomes from 20 specimens that were classified as positive by mNGS. Phylogenetic analysis revealed that 10/11 specimens from British Columbia had a closest relative to another British Columbian specimen. We found 100% concordance between phylogenetic lineage assignment and Variant of Concern (VOC) PCR results. Our assay was able to distinguish between the Alpha and Gamma variants, which was not possible with the current standard VOC PCR being used in British Columbia.

**Funding:** CN received funding from the Berkeley Existential Risk Initiative to perform this research. The funding institution played no role in the design of this study. URL: https://existence.org/.

**Competing interests:** SDC holds equity in BugSeq Bioinformatics Inc. This does not alter our adherence to PLOS ONE policies on sharing data and materials.

## Conclusions

This study supports future work examining the broader feasibility of nanopore mNGS as a diagnostic strategy for the detection and characterization of viral pathogens.

## Introduction

The global COVID-19 pandemic and ensuing public health emergency has underscored the need for rapid, comprehensive, and cost-effective viral testing strategies to respond effectively to outbreaks and implement public health policy. COVID-19 disease is caused by severe acute respiratory syndrome coronavirus 2 (SARS-CoV-2); a positive-sense RNA virus from the family *Coronaviridae* [1, 2]. The current standard for the diagnosis of many viral infections, including SARS-CoV-2, is based on real-time qualitative reverse transcription polymerase chain reaction (RT-PCR) assays [3]. Due to its low cost, reliability, and ability to diagnose infection known pathogens, RT-PCR has been at the forefront of viral diagnostics before and during the COVID-19 pandemic [4]. However, this method still requires many hours of hands-on time by skilled laboratory technicians and is limited in that it only detects a predetermined number of pathogens that its primers are designed to identify; unknown or unexpected infectious agents will be missed [5]. This is a strong rationale for exploring alternative diagnostic strategies that can detect known and novel pathogens.

Metagenomic next generation sequencing (mNGS) allows all genetic material recovered directly from a sample to be sequenced and analyzed in a culture-free manner. Sequence-independent single primer amplification (SISPA) [6] is one such mNGS approach. SISPA enables non-selective reverse transcription of all extracted RNA in a sample into cDNA and amplifies the reverse transcribed cDNA using random nonamers tagged to a known primer sequence. This method has been successfully used to detect and assemble genomes of avian RNA viruses [7], canine distemper virus [8], human enterovirus [9], chikungunya virus, Ebola virus, hepatitis C virus [10], influenza virus [11], as well as for detection of SARS-CoV-2 in a small number of samples [12, 13]. Therefore, there is a strong justification for using this approach to enable detection of pathogenic agents in diagnostic laboratories.

SISPA and mNGS have several clear advantages over targeted molecular approaches. mNGS enables detection of multiple pathogens and co-infection in a clinical sample, as well as potentially providing information on partial or full genome sequence, genotype, antimicrobial resistance, virulence, and microbiota-associated dysbiosis at a particular body site [14, 15]. Despite the potential advantages of this approach for clinical applications, mNGS techniques have not yet been widely adopted due to their high-cost, time-intensive sample preparation, limited access to sequencing infrastructure and lack of robust, easy-to-use and interpret bioinformatics systems [14]. Furthermore, the FDA has provided no specific requirements for validation of mNGS-based diagnostic assays; which has made validation and translation of mNGS tools for detection of microorganisms challenging for routine clinical microbiology laboratories [15].

The Oxford Nanopore Technologies' (ONT) MinION sequencing platform provides a method for high-throughput, and cost-effective long-read sequencing in a device that fits in the palm of a hand. Sequencing on the MinION device is also less time-intensive than the Illumina sequencing platform [14, 16]. The portability and cost-effectiveness of MinION sequencing makes Nanopore mNGS uniquely tailored for clinical applications. Despite these advances in long-read clinical sequencing applications, the field of nanopore clinical metagenomics has

been largely unexplored. To date, there are only a few studies that examine the use of nanopore-based metagenomics for clinical applications [10, 11, 17, 18].

Bioinformatic analysis is also a considerable barrier to adoption of mNGS for clinical diagnostics. The majority of available tools require command line knowledge, significant computing infrastructure, and experience translating bioinformatic results into actionable results [15, 19]. As well, traditional short read analysis services, such as One Codex and IDseq, were not designed or evaluated with third-generation data [20, 21]. Several tools have been developed recently to facilitate analysis specifically of nanopore mNGS data, including BugSeq and EPI2ME [22, https://epi2me.nanoporetech.com]. BugSeq is a bioinformatics solution designed for clinical microbiology labs, enabling the end-to-end analysis of nanopore sequencing data with a graphical user interface and cloud-based data processing. Its analytical method has been shown to have superior sensitivity and specificity compared to EPI2ME [22], and will be the primary analysis pipeline used in this study.

In this pilot study, we examine the feasibility and performance of a SISPA-based nanopore mNGS assay to detect and characterize SARS-CoV-2 from two distinct study populations using the MinION sequencing device. We aim to quantify the sensitivity and specificity of this assay for detection of SARS-CoV-2 from clinical samples. Additionally, we aim to assess the utility of this assay to generate high-quality SARS-CoV-2 genomes to be used for phylogenetic analyses and lineage determination.

## Materials & methods

### Study population and specimen collection

Clinical specimens were collected from two different populations. First, oropharyngeal swabs were collected in 2 mL of a guanidinium-based inactivation agent (Prestige Diagnostics) as part of a study conducted to estimate SARS-CoV-2 infection prevalence in a UK community from April 20 to June 17, 2020. Swab samples from 2714 individuals from around the greater Oxford area were collected to compare PCR, serology, and nanopore sequencing for SARS-CoV-2 infected versus uninfected subjects. A set of eight SARS-CoV-2 PCR positives or indeterminate samples from this population were included in the current study. Second, nasopharyngeal swab specimens collected in 3 mL viral transport medium (Yocon Bio-technology Co. Ltd) were obtained from routine SARS-CoV-2 community testing at Vancouver General Hospital (VGH) or the BC Centre for Disease Control (BCCDC) (Vancouver, British Columbia, Canada) (n = 35). RT-PCR testing for COVID-19 was performed for all samples at either the BCCDC Public Health Laboratory or the medical microbiology laboratory at VGH using either the Roche MagNA Pure extraction system (Roche Diagnostics, Laval, Canada) in combination with detection of E-gene and RdRp gene targets, or the Panther Fusion SARS-CoV-2 assay (Hologic Inc., San Diego, CA) detecting two targets in ORF1ab. Primers for the SARS-CoV-2 RNA-dependent RNA polymerase (RdRp) were developed in-house by the BCCDC Public health laboratory and primers for the E gene were based on the World Health Organization RT-qPCR protocol [3]. The human RNaseP gene was used as an internal control as suggested by the World Health Organization (https://www.who.int/csr/resources/publications/swineflu/CDCRealtimeRTPCR_SwineH1Assay-2009_20090430.pdf?ua=1). A table containing primers and probes used for these assays can be found in S1 Table. Additionally, PCR screening for potential variants of concern (VOCs) (Ex. Alpha, Beta, Gamma, Delta variants) was performed on 11 of the positive swabs obtained from VGH that were collected from May 24–26 2021. Primers and probes were designed to target the N501Y and E484K mutations (S2 Table). Swabs were stored at either -80˚C for the oropharyngeal swabs or -20˚C for the

nasopharyngeal swabs. Specimens were chosen to obtain test performance metrics for nanopore mNGS across a range of $C_t$ values (S1 Fig).

## RNA extractions

Prior to extraction, samples were vortexed and 200 μL of each sample was centrifuged at 16,000$g$ for 3 minutes to pellet host cells. 140 μL of supernatant was aspirated and viral RNA was extracted from the supernatant using the QIAmp Viral RNA kit (Qiagen) as previously described [11], and eluted in 30 μL nuclease-free water. Samples were treated with TURBO DNase (Thermo Fisher Scientific) and incubated at 37˚C for 30 minutes, followed by concentration and clean-up with the RNA Clean & Concentrator-5 kit (Zymo Research); finally, eluting in 8 μL nuclease-free water.

## SISPA amplification

SISPA amplification was performed as described previously [9–13]. Briefly, concentrated RNA was incubated with primer A (100 pmol/μL; 5′—GTTTCCCACTGGAGGATA(N₉) – 3′) and then reverse transcribed using SuperScript IV Reverse Transcriptase (Thermo Fisher Scientific). Second strand synthesis was performed using Sequenase Version 2.0 (Thermo Fisher Scientific), following which, RNase H was performed to digest any remaining RNA. Random amplification was performed on each using AccuTaq LA DNA polymerase (Thermo Fisher Scientific) and SISPA primer B (5′—GTTTCCCACTGGAGGATA—3′). This reaction underwent PCR using the following conditions: initial denaturation for 30 seconds at 98˚C, followed by 30 cycles of 94˚C for 15 seconds, 50˚C for 20 seconds, and 68˚C for 2 minutes. A final elongation step of 68˚C for 10 minutes was added, prior to a final hold at 4˚C. Amplified cDNA was purified using a 1:1 ratio of PCR Clean DX beads (Aline Biosciences) and eluted in 50 μL nuclease-free water. Amplified cDNA was quantified using a Qubit 4 Fluorometer (Thermo Fisher Scientific) and fragment lengths were assessed using the TapeStation 2200 automated electrophoresis platform (Agilent).

## Library preparation and MinION sequencing

Library preparation was performed using ONT's ligation sequencing kit (SQK-LSK109 or SQK-LSK110). Multiplexing was performed using the native barcoding expansion 96 kit (EXP-NBD196). Library preparation was performed according to the manufacturer's instructions, with several key modifications. DNA repair and end-prep were performed with 1000 fmol of input cDNA and the incubation times were increased to 30 minutes at 20˚C, followed by 30 minutes at 65˚C. For the barcoding reaction 200 fmol of input cDNA was incubated with the native barcodes and Blunt/TA Ligase Master Mix (New England Biolabs) for 20 minutes at room temperature (15–25˚C), followed by 10 minutes at 65˚C to improve barcode ligation efficiency with smaller fragments. Up to four clinical samples (90 fmol/sample) were multiplexed on each minION flowcell, with the addition of a blank viral transport medium negative control sample to each pooled library. Samples were sequenced on FLO-MIN106 flowcells on MinION MK1b sequencing devices for 72 hours using MinKNOW (Version 4.2.8, Oxford Nanopore Technologies) with live basecalling disabled.

## Sequence data analysis

Raw fast5 files were basecalled using Guppy (Version 5.0.7, Oxford Nanopore Technologies) using the—*device cuda:0* flag to enable GPU basecalling. Output fastq files were uploaded to BugSeq (version 1.1, database version: RefSeq on Jan 28, 2021) for metagenomic classification

[22], and results classification results were visualized in Recentrifuge [23]. A representative html file containing an example visualization output can be found in the S1 File. In brief, reads were demultiplexed with qcat using default run parameters (enforcing barcodes on both ends, which we have defined as stringent demultiplexing), followed by quality control with prinseq-lite. Reads shorter than 100bp or those deemed low quality (DUST score less than 7) were discarded. Reads were then classified against all of the microbial genomes in RefSeq, as well as the human genome and a library of common contaminants [see 22 for details]. Reads classified as SARS-CoV-2 were extracted and used to build a consensus sequence with Medaka. Bases with less than 20X coverage were masked in accordance with public SARS-CoV-2 sequencing guidelines (https://www.aphl.org/programs/preparedness/Crisis-Management/Documents/APHL-SARS-CoV-2-Sequencing.pdf). SARS-CoV-2 lineages were assessed using Pangolin (Version 3.1.5, github.com/cov-lineages/pangolin), and phylogenetic analysis was performed with UShER [24] (Database: GISAID, GenBank, COG-UK and CNCB [2021-07-11]). Phylogenetic trees were constructed using augur [25], rooted at the SARS-CoV-2 reference sequence, and visualized in iTOL [26]. Antimicrobial resistance genes were detected by aligning reads against the Resfinder database [27] with minimap2, disabling secondary alignments. Analysis from BugSeq outputs and visualizations were performed in RStudio (R version 4.1.0) and Python, with all code available at https://gitlab.com/bugseq/sars-cov-2-nanopore-mngs-performance [28].

## Ethics approval

This study obtained research ethics board approval from the University of British Columbia (H20-02152). Approval for collection of participant data was obtained by the Central University Research Ethics Committee at the University of Oxford (R69035). Specimens collected as part of routine testing at VGH and the BCCDC were de-identified and only contained a sample ID number, collection date, $C_t$, and VOC screening result.

# Results

## Sequence data & sample descriptions

Amplified cDNA from a total of 43 patient swabs were sequenced on MinION sequencing devices. Of these samples, 38 were either positive or had indeterminate results based on SARS-CoV-2 RT-PCR and 5 samples had negative RT-PCR results. The 38 positive and indeterminate samples had a mean $C_t$ value of 27.6 and ranged from 14.7–38.7 (S1 Fig). Sample collection dates, sample type, total read counts, as well as dual barcode reads, percent human reads, and SARS-CoV-2 reads per million reads sequenced (RPM) are present in Table 1. On average, negative controls exhibited a 29.7-fold decrease in dual barcode reads compared to the average number of dual barcode specimen reads (Mean dual barcode reads = 20,013, Q1: 158.5, Q3: 17556). SARS-CoV-2 was detected in similar abundance across our six positive control samples obtained from cultured SARS-CoV-2 viral particles (Mean RPM Dual Barcode: 103,521 ± 21,070).

## Sensitivity, specificity, & limit of detection

We evaluated the test performance of our mNGS assay for detecting SARS-CoV-2. A sample was considered positive if one or more reads were assigned to SARS-CoV-2. Across all clinical samples, we detect SARS-CoV-2 with 78.4% (95%CI 62.8%-88.6%) sensitivity and 100% specificity (95%CI 56.6%-100%) (Table 2). Previous literature has demonstrated decreased sensitivity of mNGS assays above $C_t$ 30 for other viruses [11, 29]. To assess the dependence of the

**Table 1. Study sample descriptions and sequencing results.**

| Study ID | Collection Location | Swab Type | Collection Date | Ct Value | Gene | Kit | Reads | Dual Barcode | % Human | RPM (Dual Barcode) |
|---|---|---|---|---|---|---|---|---|---|---|
| P1 | VGH | NPS | Fall 2020 | 37.1 | ORF1ab | SQK-LSK109 | 2,592,365 | 580,829 | 90 | 3,030.15 |
| P2 | VGH | NPS | Fall 2020 | 24.1 | ORF1ab | SQK-LSK109 | 2,196,488 | 425,936 | 50 | 62,401.39 |
| P3 | VGH | NPS | Fall 2020 | 14.7 | ORF1ab | SQK-LSK109 | 1,480,039 | 268,331 | 8 | 889,826.37 |
| P4 | Oxford | OPS | Spring 2020 | 25.4 | E-gene | SQK-LSK109 | 1,681,970 | 194,567 | 62 | 3,135.17 |
| P5 | Oxford | OPS | Spring 2020 | 29.9 | E-gene | SQK-LSK109 | 1,487,346 | 369,374 | 81 | 2.71 |
| P6 | Oxford | OPS | Spring 2020 | 34.1 | E-gene | SQK-LSK109 | 1,484,871 | 224,739 | 39 | 0 |
| P7 | Oxford | OPS | Spring 2020 | 35.4 | E-gene | SQK-LSK109 | 1,871,165 | 315,162 | 88 | 0 |
| P8 | Oxford | OPS | Spring 2020 | 38.7 | E-gene | SQK-LSK109 | 4,892,596 | 1,648,997 | 79 | 0 |
| P9 | Oxford | OPS | Spring 2020 | 31.7 | E-gene | SQK-LSK109 | 3,095,244 | 853,190 | 54 | 0 |
| P10 | Oxford | OPS | Spring 2020 | Indeterminate | E-gene | SQK-LSK109 | 3,195,376 | 1,209,061 | 65 | 0 |
| P11 | Oxford | OPS | Spring 2020 | Indeterminate | E-gene | SQK-LSK109 | 2,642,491 | 758,333 | 28 | 0 |
| P12 | BCCDC | NPS | Fall 2020 | 36.13 | E-gene | SQK-LSK110 | 1,894,335 | 425,729 | 91 | 2.35 |
| P13 | BCCDC | NPS | Fall 2020 | 35.21 | E-gene | SQK-LSK110 | 2,612,555 | 636,570 | 0.2 | 0 |
| P14 | BCCDC | NPS | Fall 2020 | 33.33 | E-gene | SQK-LSK110 | 3,335,378 | 794,876 | 16 | 1.26 |
| P15 | BCCDC | NPS | Fall 2020 | 33.73 | E-gene | SQK-LSK110 | 3,689,514 | 897,886 | 98 | 0 |
| P16 | BCCDC | NPS | Fall 2020 | 33.63 | E-gene | SQK-LSK110 | 2,301,355 | 593,209 | 80 | 5.06 |
| P17 | BCCDC | NPS | Fall 2020 | Indeterminate | NA | SQK-LSK110 | 1,412,609 | 384,971 | 10 | 38.96 |
| P18 | BCCDC | NPS | Fall 2020 | Indeterminate | NA | SQK-LSK110 | 1,269,020 | 256,134 | 92 | 0 |
| P19 | BCCDC | NPS | Fall 2020 | 36.33 | E-gene | SQK-LSK110 | 2,588,988 | 744,812 | 82 | 0 |
| P20 | VGH | NPS | Spring 2021 | 35.6 | ORF1ab | SQK-LSK110 | 1,535,450 | 431,421 | 48 | 2.32 |
| P21 | VGH | NPS | Spring 2021 | 34.3 | ORF1ab | SQK-LSK110 | 1,553,510 | 411,279 | 37 | 2.43 |
| P22 | VGH | NPS | Spring 2021 | 33.7 | ORF1ab | SQK-LSK110 | 1,206,439 | 328,369 | 47 | 3.05 |
| P23 | VGH | NPS | Spring 2021 | 21.4 | ORF1ab | SQK-LSK110 | 1,584,504 | 499,025 | 7 | 17,462.05 |
| P24 | VGH | NPS | 25 May, 2021 | 15.5 | ORF1ab | SQK-LSK110 | 2,875,078 | 728,905 | 84 | 58,192.77 |
| P25 | VGH | NPS | 25 May, 2021 | 16.1 | ORF1ab | SQK-LSK110 | 2,184,440 | 484,358 | 87 | 68,748.74 |
| P26 | VGH | NPS | 25 May, 2021 | 16.1 | ORF1ab | SQK-LSK110 | 968,712 | 301,091 | 49 | 493,422.25 |
| P27 | VGH | NPS | 25 May, 2021 | 17 | ORF1ab | SQK-LSK110 | 2,550,631 | 737,603 | 81 | 60,411.90 |
| P28 | VGH | NPS | 25 May, 2021 | 17.7 | ORF1ab | SQK-LSK110 | 2,151,872 | 503,298 | 87 | 22,088.31 |
| P29 | VGH | NPS | 24 May, 2021 | 20 | ORF1ab | SQK-LSK110 | 993,047 | 212,823 | 77 | 47,057.88 |
| P30 | VGH | NPS | 7 Dec, 2020 | 22 | E-gene | SQK-LSK110 | 707,288 | 253,025 | 81 | 171,129.34 |
| P31 | VGH | NPS | 26 May, 2021 | 22.8 | ORF1ab | SQK-LSK110 | 2,009,926 | 456,803 | 98 | 1,136.16 |
| P32 | VGH | NPS | 25 May, 2021 | 23.5 | ORF1ab | SQK-LSK110 | 3,173,498 | 687,623 | 99 | 373.75 |
| P33 | VGH | NPS | 26 May, 2021 | 24.4 | ORF1ab | SQK-LSK110 | 1,597,376 | 239,837 | 85 | 1,054.88 |
| P34 | VGH | NPS | 25 May, 2021 | 25.5 | ORF1ab | SQK-LSK110 | 1,103,117 | 283,010 | 99 | 38.87 |
| P35 | VGH | NPS | 25 May, 2021 | 27.3 | ORF1ab | SQK-LSK110 | 3,325,162 | 1,042,089 | 95 | 2.88 |
| P36 | VGH | NPS | 25 May, 2021 | 27.7 | ORF1ab | SQK-LSK110 | 1,374,869 | 322,646 | 87 | 27.89 |
| P37 | VGH | NPS | 20 July, 2020 | 28 | E-gene | SQK-LSK110 | 1,365,733 | 278,532 | 86 | 240.55 |
| P38 | VGH | NPS | 25 May, 2021 | 30.6 | ORF1ab | SQK-LSK110 | 4,458,073 | 1,335,187 | 98 | 49.43 |
| N1 | VGH | NPS | 23 May, 2021 | NA | NA | SQK-LSK110 | 1,803,891 | 521,584 | 96 | 0 |
| N2 | VGH | NPS | 24 May, 2021 | NA | NA | SQK-LSK110 | 1,932,656 | 645,041 | 96 | 0 |
| N3 | VGH | NPS | 24 May, 2021 | NA | NA | SQK-LSK110 | 3,421,518 | 1,053,199 | 98 | 0 |
| N4 | VGH | NPS | 23 May, 2021 | NA | NA | SQK-LSK110 | 4,947,322 | 1,539,940 | 75 | 0 |
| N5 | VGH | NPS | 23 May, 2021 | NA | NA | SQK-LSK110 | 1,386,059 | 722,140 | 90 | 0 |

**Table 2. Overall sample classification, before adjustment for barcode crosstalk.**

| | | Positive by mNGS | Negative by mNGS | Sum |
|---|---|---|---|---|
| True positive | Ct≤30 | 21 | 0 | 21 |
| | Ct30-38.7 | 8 | 8 | 16 |
| True negative | | 0 | 5 | 5 |
| Sum | | 29 | 13 | |

mNGS assay on $C_t$ value, we performed a subgroup analysis on samples above and below SARS-CoV-2 $C_t$ 30. For samples with SARS-CoV-2 $C_t < 30$, sensitivity was 100% (95%CI 84.5%-100%), while for samples with SARS-CoV-2 $C_t$ greater than 30, sensitivity was 50% (95%CI 27.8%-72.0%).

We note that two of 11 negative control samples had a single read assigned to SARS-CoV-2. We investigated these reads (further denoted as read one and two) to identify reasons for false positivity. Both reads had the expected barcode on both ends of the read as identified by BLAST. The first read exhibited 100% identity over the 24 nucleotide barcode on both ends, and the second read had 100% and 83% identity over the 24 nucleotide barcode on both ends. We next search these reads against the NCBI nucleotide database using megaBLAST to assess whether a BugSeq classification error occurred. However, both reads had top hits that exclusively matched SARS-CoV-2 with greater than 95% identity over more than 90% of their total length (923 and 1752 bases, respectively). SARS-CoV-2 was detected, despite strict dual barcode demultiplexing and removal reads with improper barcode insertions. Previous studies have identified barcode crosstalk, ranging from 0.2% to 0.3% of total classified reads, on nanopore MinION flowcell results [30, 31]. When we examined the total SARS-CoV-2 read counts for a given flowcell on flowcells with false positive negative controls, we saw that one of those flowcells has the highest total SARS-CoV-2 read count of all flowcells in this study, therefore, we would expect higher levels of barcode crosstalk for that flowcell (S2 Fig).

We adjusted for barcode crosstalk by controlling for the total number of dual-barcoded SARS-CoV-2 reads on each flowcell. If we assume 0.2% of reads have incorrect barcodes ligated on both ends, and that these misclassified reads are evenly distributed across all barcodes on the flowcell, we can subtract the estimated number of misclassified reads from each sample. This correction yielded an acceptable threshold for classifying specimens as positive or negative. After adjusting for barcode crosstalk in this manner, we find that seven samples and two negative controls with SARS-CoV-2 reads detected would be re-classified as negative, and all negative controls are therefore classified correctly. The overall sensitivity and specificity on clinical samples after adjusting for barcode crosstalk are estimated to be 59.5% (95%CI 43.5%-73.7%) and 100% (56.6%-100%), respectively. Grouping by $C_t$ value, the sensitivity estimates are 95.2% (95%CI 77.3%-99.2%) and 12.5% (95%CI 3.5%-36.0%%) for samples below and above $C_t$ 30, respectively (Table 3).

**Table 3. Overall sample classification, after adjustment for barcode crosstalk.**

| | | Positive by mNGS | Negative by mNGS | Sum |
|---|---|---|---|---|
| True positive | Ct≤30 | 20 | 1 | 21 |
| | Ct30-38.7 | 2 | 14 | 16 |
| True negative | | 0 | 5 | 5 |
| Sum | | 22 | 20 | |

### RT-qPCR/SISPA correlation, genome coverage, & SARS-CoV-2 phylogeny

We assessed the relationship between SARS-CoV-2 RT-PCR $C_t$ value and SARS-CoV-2 RPM for dual barcode reads, using stringent demultiplexing analysis parameters. SARS-CoV-2 log-RPM showed a strong linear association with RT-qPCR $C_t$ value ($R^2 = 0.71$), with lower $C_t$ values having a higher RPM on average (Fig 1). This relationship did not differ by RT-qPCR gene target (E-gene, ORF1ab, or RdRp) (S3 Fig). SARS-CoV-2 genome coverage depth showed a similar relationship, with decreasing coverage depth across the entire genome being associated with increasing $C_t$ value (Fig 2 and Table 4). We produced logistic regression models to assess the probability of attaining greater than 95% genome coverage at 1X, 20X, or 50X depth of coverage. We found that for every one unit increase in $C_t$ value, the odds of recovering a 95% complete genome were 0.765 (95% CI: 0.519, 0.961), 0.263 (95% CI: 0.023, 0.666), or 0.263 (95% CI: 0.023, 0.666) on average for coverage depths of 1X, 20X, or 50X, respectively (Fig 3). Interestingly, we did not see any difference in the likelihood of obtaining 95% coverage for 20X or 50X, despite slight differences in coverage depth for both of these thresholds (Table 4 and Fig 3).

SARS-CoV-2 metagenomic reads were used to reconstruct viral genomes. We produced ten complete (greater than 95% unambiguous bases) and one near-complete consensus genome sequence (greater than 80% unambiguous bases) from our 20 SISPA-positive clinical specimens, masking any bases with less than 20X coverage. Two partial viral genomes were constructed with 20–25% unambiguous bases. Pangolin lineage assignment was successful to all complete or near complete genomes; of these five underwent SARS-CoV-2 VOC PCR testing. All five whole or partial viral genomes were classified as SARS-CoV-2 lineages concordant with PCR results (Table 5). We also detected an additional VOC in a sample that did not undergo VOC PCR testing. We also assessed our complete or near-complete genomes in the context of global SARS-CoV-2 transmission by placing them in a phylogenetic tree containing over two million publicly available SARS-CoV-2 genomes. The ten complete genomes could be placed in the global phylogeny with high confidence (only one maximally parsimonious placement), and the near-complete genome could be placed with lower confidence (nine maximally parsimonious placements). For ten of 11 genomes derived from metagenomic data, the nearest neighbor in this tree was a genome derived from the same province of sample collection, British Columbia. Additionally, for 9/11 study genomes, 80% or more of the nearest 50 genomes were derived from British Columbia; for the remaining two study genomes, 90% or more of the nearest 50 genomes were derived from Canada (Fig 4 and S4 Fig). The UK samples did not yield well covered genomes. Subtrees with nearest neighbors for all study samples are available in the S4 Fig.

### Universal microbial detection & antimicrobial resistance

We searched the BugSeq metagenomic output of our clinical specimens for alternative respiratory viruses or viral or bacterial co-infections. We did not identify any other pathogenic viruses or atypical bacteria such as *Chlamydia pneumoniae* or *Mycoplasma pneumoniae*. We did identify several members of the normal nasopharyngeal microbiota, which when found in the lower respiratory tract, may cause disease; these included two samples with *Moraxella catarrhalis*, seven samples with *Haemophilus influenzae* or *Haemophilus parainfluenzae*, three samples with *Neisseria meningitidis*, three samples with *Staphylococcus aureus*, two samples with *Streptococcus pneumoniae* and two samples with *Klebsiella pneumoniae* (S3 Table). These results are consistent with other metagenomic sequencing results from the nasopharynx [29]. We searched our data for genes conferring antimicrobial resistance, and identified 10 genes across 6 samples. We found two beta-lactamases in our dataset: *blaTEM-234*, a class A beta-

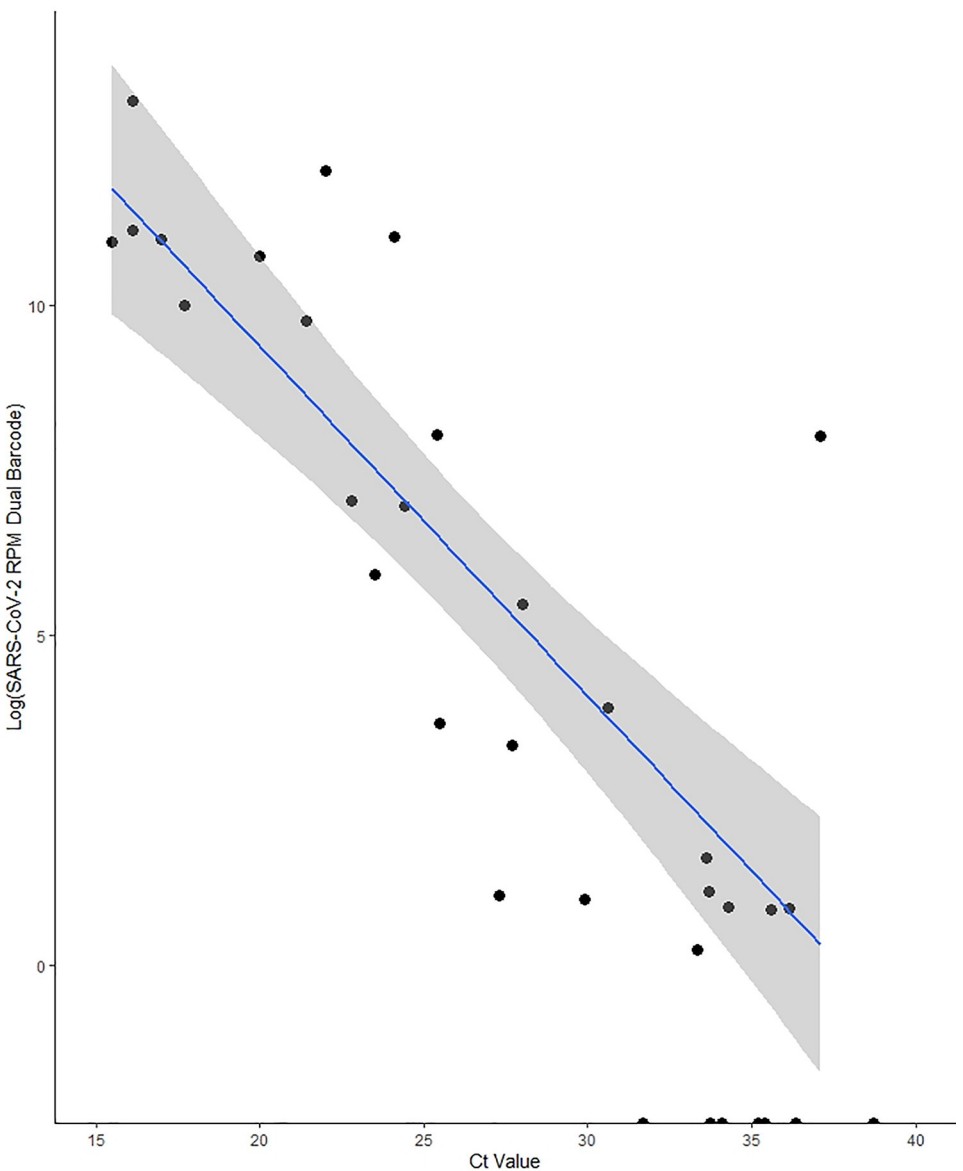

**Fig 1. Log SARS-CoV-2 reads per million reads sequenced across $C_t$ value (E gene or ORF1ab) for all RT-qPCR positive samples.** 95% confidence intervals for the linear regression line are shaded in grey. Coefficient of determination = 0.71.

lactamase which has undetermined spectrum and derived from *Escherichia coli* in sample P22, as well as *blaOXA-85*, which confers resistance to amoxicillin and amoxicillin-clavulanate, that derived from *Fusobacterium psuedoperiodonticum* (P9).

## Discussion

Here, we present a robust analysis detailing the performance of SISPA coupled with nanopore mNGS to detect and characterize SARS-CoV-2 from clinical samples. Clinical specimens exhibiting a $C_t < 30$ performed well. However, test performance declined in specimens exhibiting a $C_t \geq 30$ from 96.3% sensitivity for samples below $C_t$ 30 to 12.5% for samples above this

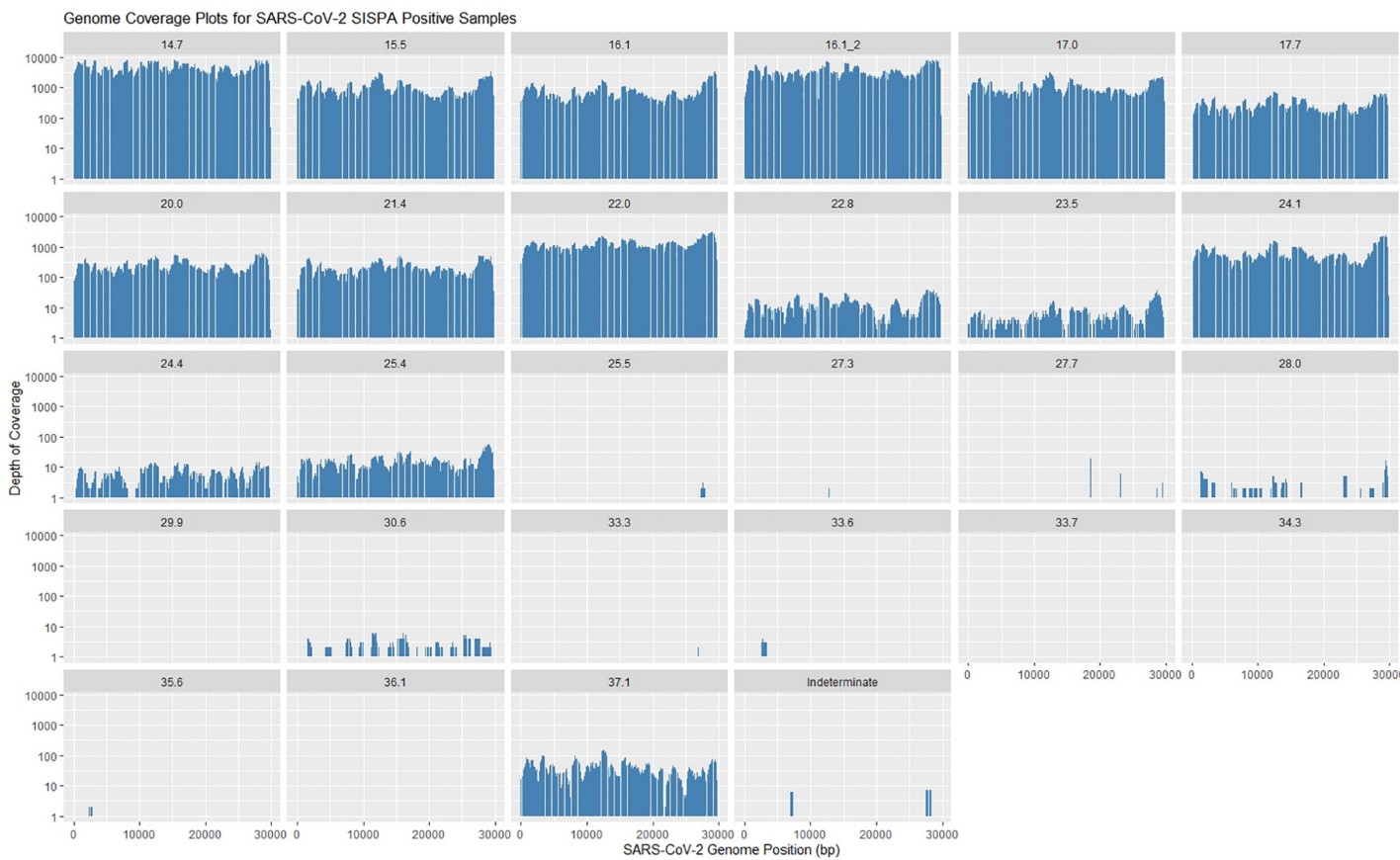

**Fig 2. Coverage depth for samples classified as positive by our classifier with log depth of coverage on the y-axis and SARS-CoV-2 reference genome position on the x-axis.**

cycle threshold. We found an exponentially declining relationship between RPM and $C_t$ value, such that the instantaneous change in read performance was fixed as illustrated in the linear relationship between Log(RPM) and $C_t$ value (Fig 1). This finding is consistent with other reports on the use of SISPA and nanopore mNGS for respiratory infections [11, 32]. However, our results are not consistent with SISPA and mNGS results from blood and serum viral diagnostics, where $C_t$ value did not drastically impact genome coverage [33]. These inconsistent results may have been influenced by sample type, sample preparation and the relative abundance of host nucleic acid in different sample types.

Despite limitations in SISPA and nanopore metagenomic sequencing sensitivity, this approach remains a valuable technique for the detection of pathogens that are novel, unexpected or uncharacterized, and therefore unsuitable for targeted approaches such as RT-qPCR or emerging CRISPR-Cas-based diagnostics, which focus only on known pathogens [34]. Unlike these existing diagnostic methods, nanopore mNGS can theoretically detect any pathogen and co-infections, characterize changes in the site-specific microbiota, and capture the carriage of critical virulence or antibiotic-resistant organisms or genes, all of which can impact patient outcomes. Our approach identified several organisms in the nasopharyngeal microbiota that may cause disease in the lower respiratory tract, consistent with sequencing results from a recent study [29]. We also did not detect any viral or atypical bacterial co-infections (S3 Table), concordant with previous reports of a low prevalence of respiratory co-infection in

**Table 4. Percent SARS-CoV-2 genome coverage for samples classified as mNGS SARS-CoV-2 positive following 0.2% crosstalk correction.**

| Study ID | Ct Value | RPM (Dual Barcode) | 50X Coverage | 20X Coverage | 1X Coverage |
|---|---|---|---|---|---|
| P1 | 37.1 | 3,030.15 | 26.25 | 80.09 | 99.85 |
| P2 | 24.1 | 62,401.39 | 99.85 | 99.94 | 100 |
| P3 | 14.7 | 889,826.37 | 99.98 | 100 | 100 |
| P4 | 25.4 | 3,135.17 | 1.57 | 23.71 | 100 |
| P23 | 21.4 | 17,462.05 | 98.85 | 99.78 | 99.95 |
| P24 | 15.5 | 58,192.77 | 99.8 | 99.91 | 100 |
| P25 | 16.1 | 68,748.74 | 99.85 | 99.97 | 100 |
| P26 | 16.1 | 493,422.25 | 99.99 | 100 | 100 |
| P27 | 17 | 60,411.90 | 99.81 | 99.89 | 100 |
| P28 | 17.7 | 22,088.31 | 99.78 | 99.79 | 100 |
| P29 | 20 | 47,057.88 | 99.59 | 99.75 | 100 |
| P30 | 22 | 171,129.34 | 99.8 | 99.85 | 100 |
| P31 | 22.8 | 1,136.16 | 0 | 20.01 | 99.99 |
| P32 | 23.5 | 373.75 | 0 | 3.87 | 98.94 |
| P33 | 24.4 | 1,054.88 | 0 | 0 | 98.74 |
| P34 | 25.5 | 38.87 | 0 | 0 | 15.25 |
| P35 | 27.3 | 2.88 | 0 | 0 | 3.63 |
| P36 | 27.7 | 27.89 | 0 | 0.3 | 3.85 |
| P37 | 28 | 240.55 | 0 | 0.07 | 47.25 |
| P38 | 30.6 | 49.43 | 0 | 0 | 72.79 |

COVID-19 positive samples [35–37]. In support of this finding, our study regions saw a dramatic reduction in incidence of other respiratory viruses (eg., influenza and RSV) and bacterial pathogens over our collection period, thought to be secondary to public health interventions.

We additionally assessed the ability of SISPA-based mNGS to classify and assemble complete or partial SARS-CoV-2 genomes from RT-qPCR positive specimens. This method can perform dual diagnostic and molecular epidemiology functions. Reliably, we were able to assemble near-complete genomes (minimum 20X coverage) up to $C_t$ 25, underscoring the ability of this approach not only to detect emerging pathogens, but also to characterize them without *a priori* knowledge of a pathogen's genome sequence. This ability contrasts to amplicon-based sequencing methods, which require the viral sequence to develop primers [38]. We performed lineage typing on metagenomic-derived SARS-CoV-2 genomes and found perfect concordance with VOC PCR on a small subset of our samples. Moreover, with the complete and partial genomes we were able to distinguish the P.1 variant from the B.1.351 variant, which our PCR assay was unable to do, as both variants contain the E484K and N501Y mutations in their spike genes targeted by the PCR assay. Our reconstructed viral genomes were further validated through phylogenetic analyses, where 10/11 samples that were of British Columbian origin were most closely related to another British Columbia genome sequence. This highlights the potential of mNGS sequencing to be an all-in-one assay which detects and characterizes pathogens of interest in near real-time, providing critical information for clinical care, infection prevention and control and public health interventions.

This study examined the methodological feasibility and validity of nanopore mNGS. We observed false positive SARS-CoV-2 reads in our negative control samples despite meticulous laboratory preparation, including performing nucleic acid extractions in a biological safety cabinet, using freshly aliquoted reagents, decontamination of all surfaces with ethanol and RNaseZap (Thermo Fisher Scientific), and performing pre-amplification steps in a dedicated

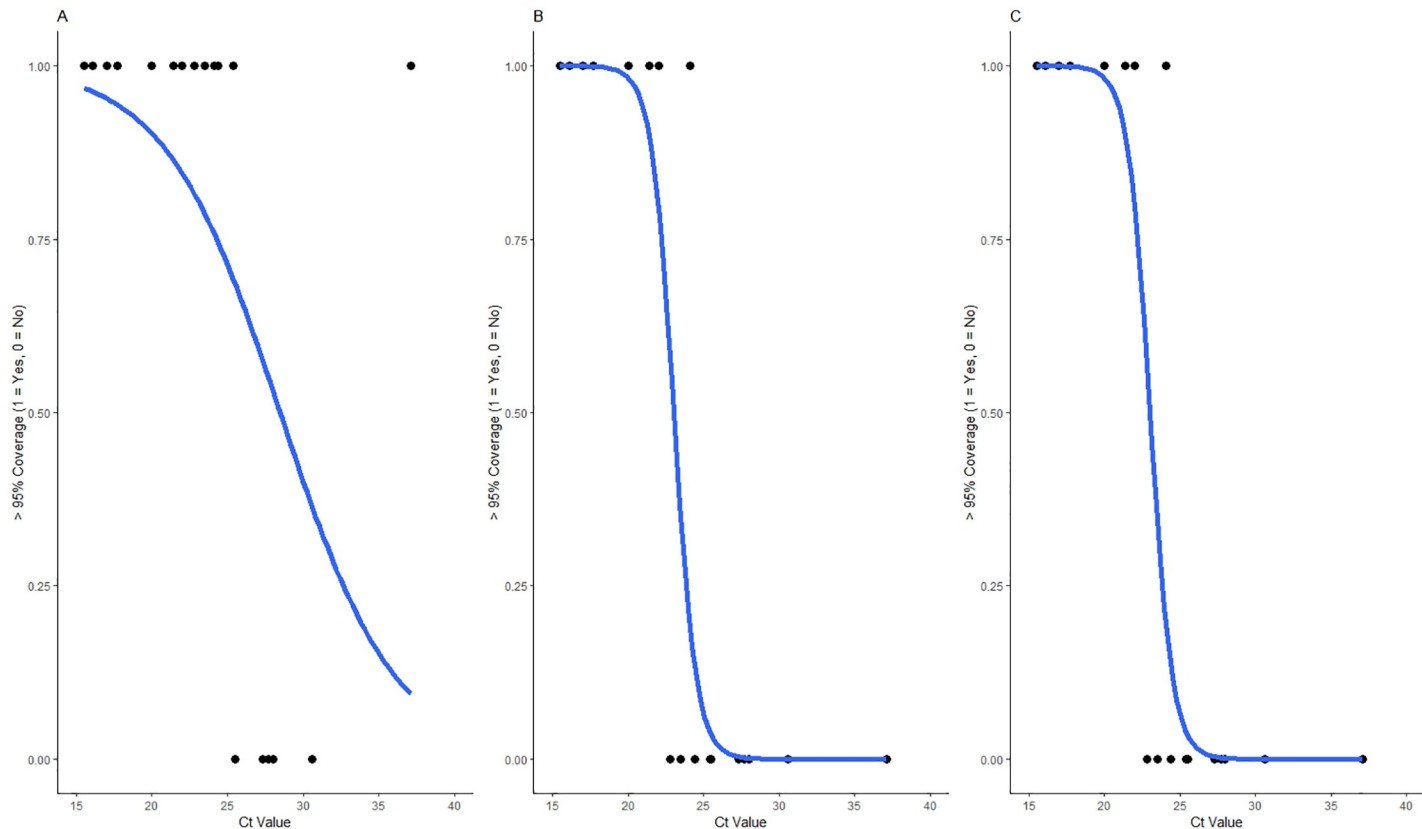

**Fig 3.** Probability of obtaining greater than 95% genome coverage (1 = Yes, 0 = No) for RT-qPCR positive study samples across $C_t$ value for **a.** 1x, **b.** 20x, and **c.** 50x genome coverage. Logistic regression models are represented in blue.

PCR area. After investigating these reads, we attribute them to barcode crosstalk, in accordance with previous studies [29, 30]. While BugSeq implements methods to minimize barcode crosstalk from nanopore sequencing, including requiring barcodes to be present at both ends of each read and removal of reads with barcodes integrated elsewhere, we developed a method to adjust the total read counts on a flowcell for barcode crosstalk. These enhancements improved assay specificity; however, sensitivity is negatively impacted by this read count adjustment. Interestingly, using the estimated 0.2% expected crosstalk between barcodes based on existing reports in the literature, we find far fewer false positive reads in our negative controls than would be expected (1 read found in each versus 3 and 107 reads expected). We do note that native barcoding on the nanopore platform is not fully optimized, leading to a significant portion of reads with only a single barcode in our sequencing datasets. This results in a decreased sensitivity, when requiring that barcodes be present on both read ends. Future advances in sequencing chemistry may reduce the prevalence of barcode crosstalk while preserving assay sensitivity.

In addition to employing automated demultiplexing and minimizing barcode crosstalk for nanopore mNGS, we validated the BugSeq as a potentially powerful clinical bioinformatics platform and workflow, including quality control, data visualization, taxonomic classification, consensus sequence generation, data aggregation, and results reporting. Although a lack of straight-forward and user-friendly bioinformatics pipelines has long been a deterrent for clinical laboratories implementing NGS and mNGS methods, our use of BugSeq as a rapid and

**Table 5. SARS-CoV-2 variant of concern PCR and Pangolin classification results.**

| Study ID | RPM (Dual Barcode) | VOC PCR Result | Pangolin Lineage (Scorpio Call) |
|---|---|---|---|
| P1 | 3,030.15 | Not Performed | B.1.2 |
| P2 | 62,401.39 | Not Performed | B.1.128 |
| P3 | 889,826.37 | Not Performed | B.1.2 |
| P4 | 3,135.17 | Not Performed | None |
| P23 | 17,462.05 | Not Performed | B.1.2 |
| P24 | 58,192.77 | Not Performed | P.1 (Gamma) |
| P25 | 68,748.74 | Presumptive Positive Variant of Concern. Spike gene N501Y and E484K mutations DETECTED by NAT. | P.1 (Gamma) |
| P26 | 493,422.25 | Presumptive Positive B.1.1.7 Variant of Concern. Spike gene N501Y mutation DETECTED by NAT. No E484K mutation detected. | B.1.1.7 (Alpha) |
| P27 | 60,411.90 | Presumptive Positive B.1.1.7 Variant of Concern. Spike gene N501Y mutation DETECTED by NAT. No E484K mutation detected. | B.1.1.7 (Alpha) |
| P28 | 22,088.31 | Presumptive Positive Variant of Concern. Spike gene N501Y and E484K mutations DETECTED by NAT. | P.1 (Gamma) |
| P29 | 47,057.88 | Presumptive Positive B.1.1.7 Variant of Concern. Spike gene N501Y mutation DETECTED by NAT. No E484K mutation detected. | B.1.1.7 (Alpha) |
| P30 | 171,129.34 | Not Performed | B.1.36.36 |
| P31 | 1,136.16 | Presumptive Positive Variant of Concern. Spike gene N501Y and E484K mutations DETECTED by NAT. | None |
| P32 | 373.75 | Negative. No Spike gene N501Y or E484K mutations detected by NAT. | None |
| P33 | 1,054.88 | Presumptive Positive B.1.1.7 Variant of Concern. Spike gene N501Y mutation DETECTED by NAT. No E484K mutation detected. | None |
| P34 | 38.87 | Presumptive Positive Variant of Concern. Spike gene N501Y and E484K mutations DETECTED by NAT. | None |
| P35 | 2.88 | Not Performed | None |
| P36 | 27.89 | Presumptive Positive Variant of Concern. Spike gene N501Y and E484K mutations DETECTED by NAT. | None |
| P37 | 240.55 | Not Performed | None |
| P38 | 49.43 | Negative. No Spike gene N501Y or E484K mutations detected by NAT. | None |

robust bioinformatics tool has demonstrated the utility of user-friendly platforms for clinical diagnostics and public health service. Indeed, other groups adopting MinION sequencers in clinical microbiology laboratories have reached similar conclusions [39, 40].

Our pilot study has several limitations. Despite the MinION sequencing device providing high-throughput sequencing data in real-time, this protocol is still significantly more time intensive than RT-PCR as a diagnostic method, requiring a minimum of 12 hours from sample collection to prepared library, and another 72 hours of sequencing to reach maximal pathogen detection sensitivity (although results could be available in as little as 1–2 hours for high viral load samples). The use of liquid handling robots for automated sample extraction, nucleic acid amplification, and library preparation may aid in standardization. Additionally, examining the feasibility of a less time intensive library preparation protocol such as the Rapid Barcoding Kit (Oxford Nanopore Technologies) for this approach will aid in the optimization and translation of nanopore mNGS for routine clinical use. The SISPA approach is also limited in that it performs random amplification of both host and microbial nucleic acids. The high percentage of host RNA in nasopharyngeal/oropharyngeal swabs limits our ability to rapidly detect viruses with comparable sensitivity to PCR, requiring deeper sequencing than what is currently feasible on a MinION. Therefore, this sequencing strategy may not be optimal for samples expected to have very few viral or bacterial nucleic acids where sensitivity is paramount. We note that

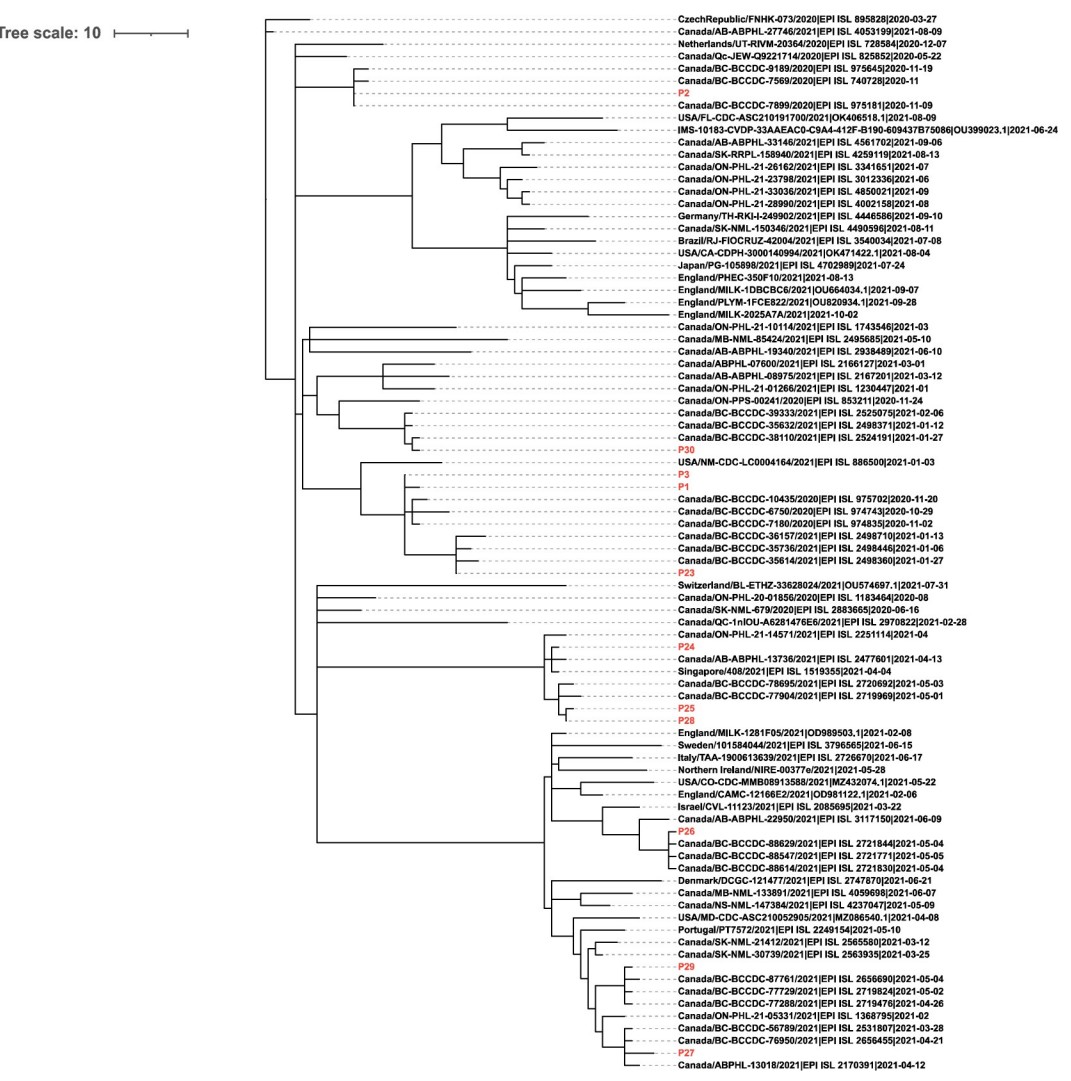

**Fig 4. Study samples (marked as P1, P2, etc.), their nearest three neighbors, and a random selection of 25 Canadian and 25 global SARS-CoV-2 sequences for context.**

while assay sensitivity has played an important role in public discourse surrounding SARS-CoV-2 testing, there is some data to suggest that lower viral loads cannot be cultured and are less likely to be transmissible (https://www.idsociety.org/globalassets/idsa/public-health/covid-19/idsa-amp-statement.pdf). This issue is further complicated by the difficulty of employing host nucleic acid removal techniques on low-biomass samples. Interestingly, the detection of host nucleic acids via mNGS may be useful, as samples with lower host nucleic acid content have been shown to be associated with higher rates of false-negative COVID-19 RT-PCR tests, presumably due to sample quality [41]. Regardless, methods to enrich for pathogen sequences or deplete host DNA to increase sensitivity have been examined [42–44], and may prove useful for future clinical metagenomics studies.

Our pilot study represents the first analysis to examine the performance and feasibility of SISPA-based nanopore mNGS for the detection and characterization of SARS-CoV-2. We improve on previous studies through quantification of the performance of this method, as well as examining the utility of this assay to assemble high-quality genomes for phylogenetic

analysis. We were able to successfully detect SARS-CoV-2 with 100% specificity and near perfect sensitivity for samples below $C_t$ 30 when adjusting for barcode crossover. We were also able to assemble SARS-CoV-2 genomes and characterize viral lineages reliably in 10/13 of samples below $C_t$ 25. This assay has the ability not only to detect known pathogens and co-infections, but can also detect emerging pathogens, assess microbiota states, and capture resistance and virulence genes. This approach holds promise as a tool for clinical diagnostics and public health surveillance.

## Supporting information

**S1 Table.** A. Primers and B. probes for RT-PCR testing performed at the BCCDC Public Health Laboratory and VGH.
(DOCX)

**S2 Table.** A. Primers, B. probes, and C. reaction conditions for Variant of Concern PCR testing.
(DOCX)

**S3 Table. Detection of potential pathogens in study samples.** Only pathogens with greater than 1% relative abundance in samples with at least 1000 bacterial reads were included.
(DOCX)

**S1 Fig. Distribution of SARS-CoV-2 RT-qPCR positive study samples across $C_t$ values.**
(DOCX)

**S2 Fig. Comparison of the total number of SARS-CoV-2 reads across all samples on a flowcell, stratified by flowcells with and without a false-positive negative control.**
(DOCX)

**S3 Fig. Scatterplot of Log(SARS-CoV-2 RPM) against $C_t$ value stratified by E-gene, ORF1ab, or RdRp.**
(DOCX)

**S4 Fig. Phylogenetic subtrees of study samples compared with 2,447,008 publicly available SARS-CoV-2 genomes.** Study samples are marked in red, publicly available genomes are black.
(DOCX)

**S1 File. Representative visualization file for BugSeq metagenomic classification results produced by Recentrifuge.**
(HTML)

## Acknowledgments

We would like to acknowledge the British Columbia Public Health Laboratory and the Medical Microbiology Laboratory at Vancouver General Hospital for providing RT-PCR testing for a subset of our samples, as well as John Tyson from the BC Centre for Disease Control for technical support and troubleshooting.

## Author Contributions

**Conceptualization:** Nick P. G. Gauthier, Cassidy Nelson, Michael B. Bonsall, Samuel D. Chorlton, Amee R. Manges.

**Data curation:** Nick P. G. Gauthier, Cassidy Nelson, Samuel D. Chorlton.

**Formal analysis:** Nick P. G. Gauthier, Samuel D. Chorlton.

**Funding acquisition:** Cassidy Nelson, Michael B. Bonsall.

**Investigation:** Nick P. G. Gauthier, Kerstin Locher, Marthe Charles, Clayton MacDonald, Samuel D. Chorlton, Amee R. Manges.

**Methodology:** Nick P. G. Gauthier, Cassidy Nelson, Kerstin Locher, Samuel D. Chorlton.

**Project administration:** Nick P. G. Gauthier, Samuel D. Chorlton, Amee R. Manges.

**Resources:** Cassidy Nelson, Michael B. Bonsall, Kerstin Locher, Mel Krajden, Samuel D. Chorlton, Amee R. Manges.

**Software:** Samuel D. Chorlton.

**Supervision:** Michael B. Bonsall, Marthe Charles, Amee R. Manges.

**Validation:** Nick P. G. Gauthier, Samuel D. Chorlton.

**Visualization:** Nick P. G. Gauthier, Samuel D. Chorlton.

**Writing – original draft:** Nick P. G. Gauthier, Samuel D. Chorlton.

**Writing – review & editing:** Nick P. G. Gauthier, Cassidy Nelson, Michael B. Bonsall, Kerstin Locher, Marthe Charles, Clayton MacDonald, Mel Krajden, Samuel D. Chorlton, Amee R. Manges.

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
