## [Decision Letter · Decision Letter 0]

19 Oct 2021

PONE-D-21-28544Nanopore Metagenomic Sequencing for Detection and Characterization of SARS-CoV-2 in Clinical SamplesPLOS ONE

Dear Dr. Manges,

Thank you for submitting your manuscript to PLOS ONE. After careful consideration, we feel that it has merit but does not fully meet PLOS ONE’s publication criteria as it currently stands. Therefore, we invite you to submit a revised version of the manuscript that addresses the points raised during the review process.

We look forward to receiving your revised manuscript.

Kind regards,

Ruslan Kalendar

Academic Editor

PLOS ONE

Journal Requirements:

[We would like to acknowledge the Berkeley Existential Risk Initiative for providing funding for this project. We would also like to acknowledge the British Columbia Public Health Laboratory and the Medical Microbiology Laboratory at Vancouver General Hospital for providing RT-PCR testing for a subset of our samples, as well as John Tyson from the BC Centre for Disease Control for technical support and troubleshooting.]

 [CN received funding from the Berkeley Existential Risk Initiative to perform this research. The funding institution played no role in the design of this study. URL: https://existence.org/]

[I have read the journal's policy and the authors of this manuscript have the following competing interests: SDC holds equity in BugSeq Bioinformatics Inc]. 

Reviewers' comments:

Reviewer's Responses to Questions

**Comments to the Author**

1. Is the manuscript technically sound, and do the data support the conclusions?

Reviewer #1: Yes

2. Has the statistical analysis been performed appropriately and rigorously? 

Reviewer #1: Yes

3. Have the authors made all data underlying the findings in their manuscript fully available?

Reviewer #1: Yes

4. Is the manuscript presented in an intelligible fashion and written in standard English?

Reviewer #1: Yes

5. Review Comments to the Author

Reviewer #1: 

Summary:

The authors present an interesting manuscript characterizing the utility of a metagenomic next-generation sequencing approach for detecting SARS-CoV-2 in clinical samples. They illustrate strong specificity and sensitivity of the assay in samples with higher viral loads (Ct < 30). The major contribution of the proposed method is the utility of the assembled genomes to distinguish SARS-CoV-2 variants as verified by Variant of Concern PCR.

The manuscript presents findings of original research that would be of interest to the wider research community. However, there are some concerns regarding the reporting quality which if addressed can significantly strengthen the manuscript.

The introduction can be expanded, specifically, the last paragraph discussing the stated goals of the manuscript.

What do the authors mean by negative and positive controls? Are these sample status based on the RT-PCR? Why are there 11 negative controls (line 241) while only 5 negative RT-PCR tests?

Also if there are two false positives, how is this not reflected in the results for the sample classification before adjusting for barcode crosstalk (Table 2)?

Restate specificity for Ct > 30 as 30 to max Ct, which is 38?

How does the nanopore SISPA based MGS compare to other MGS for SARS-CoV-2?

Figure 4 in its current format is not adding sufficient value to the manuscript. Instead it could show the phylogeny of the 10 genomes with a small subset of samples derived from BC vs other provinces to clearly illustrate the point that the genomes cluster closer with samples from BC. This is a major result of the manuscript and can be illustrated more convincingly.

There are some contradictory statements in the manuscript regarding the novelty of mNGS for SARS-CoV-2. In the introduction, the authors state that existing efforts have detected SARS-CoV-2 based on mNGS (line 73). However in discussion (line 447), they state that this study provides the first analysis of mNGS for characterizing SARS-CoV-2. Please clarify how this study compares with the prior work and offers improvement of existing methodology.

“Our assay was able to distinguish between the Alpha and Gamma variants, which was not possible with a VOC PCR.”

This is slightly misleading considering the VOC PCR is able to distinguish alpha and gamma variants, e.g. for P25 and P26 studies.

6. PLOS authors have the option to publish the peer review history of their article (what does this mean?). If published, this will include your full peer review and any attached files.

Reviewer #1: No

---

## [Author Response · Author response to Decision Letter 0]

22 Oct 2021

Academic Editor:

We have rigorously reviewed the PLOS ONE style requirements and believe that our manuscript aligns with the requirements.

[We would like to acknowledge the Berkeley Existential Risk Initiative for providing funding for this project. We would also like to acknowledge the British Columbia Public Health Laboratory and the Medical Microbiology Laboratory at Vancouver General Hospital for providing RT-PCR testing for a subset of our samples, as well as John Tyson from the BC Centre for Disease Control for technical support and troubleshooting.]

 [CN received funding from the Berkeley Existential Risk Initiative to perform this research. The funding institution played no role in the design of this study. URL: https://existence.org/]

We have amended the manuscript document to remove funding in our acknowledgements section. We have no further information to add to the funding statement. 

[I have read the journal's policy and the authors of this manuscript have the following competing interests: SDC holds equity in BugSeq Bioinformatics Inc]. 

We have updated our competing interests statement to include the statement that was included in our cover letter and have submitted this updated cover letter to the manuscript submission portal. We confirm that our competing interests align with PLOS ONE policies on sharing data and materials.

Data accession numbers are available within the manuscript and the data have been made freely available online. 

Supporting information captions have been included at the very end of the manuscript file in accordance with the supporting information guidelines. In-text citations have been reviewed and no changes have been made.

We have reviewed the reference list to ensure it is complete and correct. No revisions have been made to the reference list. 

Reviewer #1:

Summary:

The authors present an interesting manuscript characterizing the utility of a metagenomic next-generation sequencing approach for detecting SARS-CoV-2 in clinical samples. They illustrate strong specificity and sensitivity of the assay in samples with higher viral loads (Ct < 30). The major contribution of the proposed method is the utility of the assembled genomes to distinguish SARS-CoV-2 variants as verified by Variant of Concern PCR.

The manuscript presents findings of original research that would be of interest to the wider research community. However, there are some concerns regarding the reporting quality which if addressed can significantly strengthen the manuscript.

The introduction can be expanded, specifically, the last paragraph discussing the stated goals of the manuscript.

We have expanded the last paragraph of the introduction to more explicitly state the objectives and goals of our study, as well as the implications of these results to the field of diagnostic infectious diseases. 

What do the authors mean by negative and positive controls? Are these sample status based on the RT-PCR? Why are there 11 negative controls (line 241) while only 5 negative RT-PCR tests?

For each multiplexed flowcell, we ran one negative (for each of the 11 sequencing runs) and one positive control (for the first 6 flowcells, after which we were confident in our ability to successfully detect SARS-CoV-2). Our positive control was extracted SARS-CoV-2 virions from cell culture to ensure that our assay was able to detect SARS-CoV-2 reliably. The negative control we used was a blank viral transport medium control (ie. extracted and randomly amplified from a tube of viral transport medium that was sterile), which was used to ensure there was no significant contamination of our reagents or barcode crossover during sequencing. We previously described our negative control in lines 175-177 of the manuscript file. Line 225 has been updated to include more detail about our positive control. The five negative RT-PCR tests represent testing for 5 negative clinical specimens. 

so if there are two false positives, how is this not reflected in the results for the sample classification before adjusting for barcode crosstalk (Table 2)?

These two false-positive, negative controls were not reflected in table two as these did not represent clinical specimens, but rather blank controls that likely had SARS-CoV-2 reads assigned to them due to barcode crosstalk. 

Restate specificity for Ct > 30 as 30 to max Ct, which is 38?

Both Tables 2 and 3 have been updated to reflect the range of Ct values. Ct 30-38.7. Ct 38.7 was the maximum Ct value for any of our samples. 

How does the nanopore SISPA based MGS compare to other MGS for SARS-CoV-2?

Although SISPA and other MGS methods have been used for detection of SARS-CoV-2 both in nasopharyngeal swabs, as well as wastewater and fecal samples, these studies did not report test performance metrics such as sensitivity and specificity and often report solely SARS-CoV-2 reads per sample and a Ct value. Furthermore, these studies have not included data on assay controls, making comparisons or assessments of performance difficult. We suspect that mNGS protocols without an amplification step would have lower sensitivity than a random amplification-based approach such as SISPA. There is some evidence for this in Mostafa et al. (2020), where they report false negative mNGS results for samples exhibiting a Ct value as low as Ct 21.0. However, without accurate test performance data to compare these other mNGS methods to our own, we decided not to include any comparisons to other studies in our manuscript due to differences in test performance reporting.

Figure 4 in its current format is not adding sufficient value to the manuscript. Instead it could show the phylogeny of the 10 genomes with a small subset of samples derived from BC vs other provinces to clearly illustrate the point that the genomes cluster closer with samples from BC. This is a major result of the manuscript and can be illustrated more convincingly.

Figure 4 along with the figure legend has been updated to include a subset of the closest three samples related to our clinical samples (almost entirely BC samples), as well as 25 randomly sampled Canadian and 25 randomly sampled global samples. Based on these new samples, the conclusion that our samples cluster most closely with BC samples can more accurately be drawn. We believe changes to this figure more accurately highlights this result. 

There are some contradictory statements in the manuscript regarding the novelty of mNGS for SARS-CoV-2. In the introduction, the authors state that existing efforts have detected SARS-CoV-2 based on mNGS (line 73). However in discussion (line 447), they state that this study provides the first analysis of mNGS for characterizing SARS-CoV-2. Please clarify how this study compares with the prior work and offers improvement of existing methodology.

Although other studies have used mNGS to detect SARS-CoV-2 (as stated in the introduction), our work represents the first study to both, assess the performance (ie. sensitivity and specificity), as well as characterize SARS-CoV-2 genomes through phylogenetic analysis. Two previous preprints have used nanopore-based SISPA mNGS to detect SARS-CoV2 (ref 13) or to assess within host SARS-CoV-2 diversity (ref 12), but neither of these studies quantified performance metrics for their assay or demonstrated the utility of this assay for phylogenetic analysis. We have updated the introduction (lines 109-111), as well as the discussion (lines 476-479) to be clearer about how our study represents an important contribution to the literature. 

“Our assay was able to distinguish between the Alpha and Gamma variants, which was not possible with a VOC PCR.”

This is slightly misleading considering the VOC PCR is able to distinguish alpha and gamma variants, e.g. for P25 and P26 studies.

We have rephrased this statement to clarify that it only applies to the VOC PCR being used at Vancouver General Hospital for surveillance in British Columbia.

---

## [Editor Report · Decision Letter 1]

26 Oct 2021

Nanopore Metagenomic Sequencing for Detection and Characterization of SARS-CoV-2 in Clinical Samples

PONE-D-21-28544R1

Dear Dr. Manges,

We’re pleased to inform you that your manuscript has been judged scientifically suitable for publication and will be formally accepted for publication once it meets all outstanding technical requirements.

Kind regards,

Ruslan Kalendar

Academic Editor

PLOS ONE

---

## [Editor Report · Acceptance letter]

5 Nov 2021

PONE-D-21-28544R1 

Nanopore Metagenomic Sequencing for Detection and Characterization of SARS-CoV-2 in Clinical Samples 

Dear Dr. Manges:

I'm pleased to inform you that your manuscript has been deemed suitable for publication in PLOS ONE. Congratulations! Your manuscript is now with our production department. 

Kind regards, 

on behalf of

Professor Ruslan Kalendar 

Academic Editor

PLOS ONE